# Improving the Extraction of Catechins of Green Tea (*Camellia sinensis*) by Subcritical Water Extraction (SWE) Combined with Pulsed Electric Field (PEF) or Intense Pulsed Light (IPL) Pretreatment

**DOI:** 10.3390/foods10123092

**Published:** 2021-12-13

**Authors:** Hee-Jeong Hwang, Yu-Gyeong Kim, Myong-Soo Chung

**Affiliations:** 1Research Institute of Biotechnology and Medical Converged Science, Dongguk University, Seoul 10326, Korea; piatop@hanmail.net; 2Department of Food Science and Engineering, Ewha Womans University, Seoul 03760, Korea; dbrudjkljkl@naver.com

**Keywords:** subcritical water extraction, pulsed electric field, intense pulsed light, green tea, tea catechins

## Abstract

The aim of this study was to find the optimum condition of pulsed electric field (PEF) and intense pulsed light (IPL) for the enhancement of subcritical water extraction (SWE), which is an eco-friendly extraction method, for extracting tea catechins from green tea leaves (*Camellia sinensis*). The leaves were treated with PEF under conditions of electric field strength (1, 2 and 3 kV/cm) during 60 s. Moreover, IPL was applied at various voltages (800, 1000, and 1200 V) for 60 s. The SWE was performed for 5 min at varying temperatures (110, 130, 150, 170, and 190 °C). The maximum yield of total catechin was 44.35 ± 2.00 mg/g dry green tea leaves at PEF treatment conditions of 2 kV/cm during 60 s, as well as the SWE temperature of 130 °C. In the case of IPL treatment, the largest amount of total catechin was 48.06 ± 5.03 mg/g dry green tea leaves at 800 V during 60 s when the extraction temperature was 130 °C. The total catechin content was increased by 15.43% for PEF and 25.09% for IPL compared to the value of untreated leaves. This study verified that PEF and IPL had a positive effect on the enhancement of tea catechins extraction from green tea leaves using SWE.

## 1. Introduction

Green tea (*Camellia sinensis*) leaves are the most widely known and beloved tea materials due to their attractive aroma and taste. Numerous studies have been conducted to analyze and use their abilities, due to their excellent health promoting abilities, such as anti-carcinogenic effects, antihypertensive effects, and antioxidant activity [1,2,3]. In the case of dried green tea leaves, they contain 10–35% (*w*/*w*) polyphenols including catechins, phenolic acids, and glycosides [4]. Among them, tea catechins make up 80–90% of the flavonoids of green tea leaves. There are many kinds of catechins in green tea, which have different structures and bioactivity. Representatively, (–)-catechin (Ct), (–)-catechin gallate (CG), and (–)-gallocatechin gallate (GCG), and their respective epimers (–)-epicatechin (EC), (–)-epicatechin gallate (ECG), and (–)-epigallocatechin gallate (EGCG) exist (Appendix A) and the epimerization of tea catechins is reversible [5]. The catechins have antioxidant properties due to their phenolic groups, and their antioxidant capacity is known to correlate with the percentage of pyrogallol and hydroxyl groups [6]. Among them, EGCG which has the best antioxidant activity is a catechin that exists only in tea [7]. EGCG (about 59% of total catechins) is the highest amount of catechin in green tea [8], and EGCG has excellent anti-cancer, anti-inflammation ability [9]. Green tea is mainly consumed in the form of a beverage, and several factors such as the structure of green tea leaves, extraction temperature, extraction time, leaf to water ratio, agitation, stirring, and squeezing of teabag affect the content of catechins in green tea drink [10]. The content of catechin has a great effect on the positive health promotion function of green tea. However, they have a low bioavailability. In addition, the existing methods of extraction using ethanol and methanol are limited by strict legal statutes due to their toxicity and harmlessness [11]. Therefore, a technology that achieves a higher extraction concentration of catechins is needed.

Subcritical water extraction (SWE) is the method of raising the boiling point of water in a high-pressure environment (about 10 MPa) and keeping the water in a liquid state for extraction at a temperature between 100 °C (boiling point of water) and 374 °C (critical point of water) [12]. As the temperature of water rises, the polarity of water changes to nonpolar, which makes it easier to dissolve non-polar and moderately polar materials similar to organic solvents [13,14,15]. By increasing the temperature of the water at a constant pressure, the dielectric constant (ε) of the water decreases from 80 (25 °C) to 27 (225 °C), similar to the dielectric constant of the organic solvents methanol (ε = 33) and ethanol (ε = 24) [16]. Recently, research has been actively conducted on eco-friendly extraction methods that are using harmless solvents and having better extraction efficiency. In addition, SWE has been actively used as a method for extracting various functional materials from different plants. SWE was used for the extraction of antioxidant materials from rosemary leaves, onion skin, and potatoes [17,18,19]. The extraction of catechins from green tea was traditionally carried out with organic solvents. However, the extraction of black tea with 80% methanol, 70% ethanol, and water showed that the largest amount of catechin was extracted when water was used as a solvent in the previous study [10]. Therefore, tea catechins are water friendly substances and more soluble in water than organic solvents. Due to this property of catechin, a previous study using SWE for the extraction of tea catechins from green tea leaves has shown that epicatechin has high extraction efficiency in subcritical water at a low temperature, 110–190 °C [11]. This study intends to apply non-thermal techniques to extract a higher concentration of catechin from green tea leaves than previous studies.

Pulsed electric field (PEF) is a method of putting a sample in the water or solvents with two electrodes, and applying a high voltage of electric pulses for a few seconds. PEF has been mainly used as a method of non-thermal sterilization, but now it is used as an extraction method itself or as a method of improving extraction efficiency [20,21]. Many studies have mentioned that the important conditions for improving the extraction efficiency of bioactive compounds, such as flavonoids from plant cells are the electric field strength and frequency. Takaki et al. (2011) identified that the PEF treatment on grape peels induces permeabilization of the peels, thus it causes an increase of total polyphenols from 2.5 to 4.78 μmM gallic acid eq./100 g when the applied voltage of PEF was 30 to 60 kV at 5 pulses/s [22]. Moreover, Zderic et al. (2013) obtained 27% of polyphenol extraction yield when fresh tea leaves were treated with the PEF intensity of 0.9 kV/cm, while the yield was only 3% when the intensity was 0.1 kV/cm [23].

Intense pulsed light (IPL) is also one of the emerging non-thermal disinfection technologies using high intensity and short bursts of light [24]. To date, IPL is studied for the surface disinfection of food, and the industrial application is for disinfection only. However, recently, several studies confirmed that the extraction efficiency of health functional substances from plants is increased after the IPL treatment to the plant surface. For example, the extraction yields of the chlorophylls was improved up to 1.3 times after 6 J/cm^2^ of IPL treatment on the surface of fresh-cut avocados [25]. Moreover, the contents of lycopene, α-carotene, and β-carotene in tomatoes are significantly increased after 30 J/cm^2^ of IPL treatments [26]. PEF and IPL are novel non-thermal technologies with environmental benefits, such as improving the energy efficiency of the process, reducing the use of non-renewable resources, and a lower emission of air pollutants, such as CO_2_ [27].

In this study, the combined extraction processing of PEF or IPL with SWE was conducted, and verified the effectiveness of PEF and IPL as a pretreatment technology for enhancing the extraction of tea catechins from green tea leaves. Moreover, through confirming the extraction amount of catechins according to the various conditions of PEF and IPL pretreatment, and observing the surface morphology of green tea leaves after the treatments, the optimum conditions of PEF/IPL–SWE processing for maximizing extraction efficiency were resolved.

## 2. Materials and Methods

### 2.1. Green Tea Sample

The dried green tea (*Camellia sinensis*) leaves, which were harvested in Boseong, Korea, were purchased (the measured moisture content of dried leaves were below 5%). The purchased sample was kept in a place where the light was blocked and not damp at room temperature until the experiment was conducted.

### 2.2. PEF Treatment

The PEF treatment was conducted using the 5 kW pulsed generator (HVP-5, DIL, Quakenbrueck, Germany) equipped with a batch-type treatment chamber. The chamber had two parallel electrodes (10 × 5 cm^2^) and the distance between the electrodes was 8 cm. Before the PEF treatment, 20 g of dried green tea leaves were immersed in 250 mL of tap water. The PEF treatment time was 60 s and the electric field strengths were 1, 2, and 3 kV/cm. The pulse frequency was 50 Hz and the pulse width was 25 μs (for 1 and 2 kV/cm) or 20 μs (for 3 kV/cm). The total input energy for each sample was automatically calculated by integrating the electric field strength, pulsed frequency, pulse width, and treatment time in equipment. The electrical conductivity of green tea samples was measured by the conductivity meter (CP-50N, Istek, Seoul, Korea). By the measured electrical conductivity, the degree of damage by the PEF treatment was represented as the electrical conductivity disintegration index (*Z*), as shown in Equation (1) [28]. In this study, the electrical conductivity of the control sample was used as the standard of *Z* = 0, and the value of green tea leaves, which was completely frozen and dissolved, was measured for the standard of Z = 1.
Z = (σ − σ_i_)/(σ_d_ − σ_i_)(1)

Z: Electrical conductivity disintegration index.

σ: Electrical conductivity of sample.

σ_i_: Electrical conductivity of the untreated (initial).

σ_d_: Electrical conductivity of the completely damaged tissue.

### 2.3. IPL Treatment

The IPL device used in this study was self-designed and the device consists of a power supply, treatment chamber, and xenon lamp (NL4006, XAP series, Heraeus Noblelight, Cambridge, UK), which emits light at wavelengths from 200 to 1100 nm [29]. Before the IPL treatment, 20 g of dried green tea leaves were immersed in 250 mL of tap water, and then sieved to remove water as much as possible. Then, the green tea leaves were spread and placed on the plastic plates with a diameter of 14.5 cm to maximize the surface area for light exposure. The IPL treatment was applied on both sides of the green tea leaves and the IPL treatment time of each side was 60 s. In addition, the treatment voltages were 800, 1000 and 1200 V. The frequency was 5 Hz and the pulse width was 0.1 ms. The distance between the sample of green tea leaves and xenon lamp was 9 cm. The irradiance spectrum of one pulse of IPL was measured by the spectroradiometer (ILT-950, International Light Technologies, Peabody, MA, USA), and the total fluences delivered to the sample is calculated as the fluence of one pulse (J/cm^2^·pulse) × processing time (s) × frequency (Hz) [24]. The measured total fluences in this study were 9.84, 14.39, and 20.47 J/cm^2^ at 800, 1000 and 1200 V, respectively.

### 2.4. SWE Treatment

SWE was conducted by an accelerated solvent extractor (ASE 350, Dionex, Sunnyvale, CA, USA) [30]. Wet green tea leaves (2 g) were used for extraction, and one sheet of filter paper (40 grade, Whatman, Maidstone, UK) was placed on the bottom of the 22 mL stainless-steel cell. Distilled water was used as an extraction solvent. The conditions of the extraction temperature were 110, 130, 150, 170, and 190 °C and the extraction time was 5 min. The entire parameters, such as extraction time and pressure, were set based on the capacity of the equipment and the conditions of the previous research [11]. After the extraction, the extract was collected by the movement from the extraction cell to the collection vial. Extracts of all conditions were dried at −80 °C for 24 h by a freeze-dryer (Ilshin, Gyeonggi-do, Korea).

### 2.5. High-Performance Liquid Chromatography

For the analysis of six kinds of tea catechins, Ct, CG, GCG, EC, ECG, and EGCG, which are extracted from the green tea leaves, the method and standards for the analysis of six catechins from the green tea extracts used in the previous study were used with some modifications [30]. The quantitative analysis of tea catechins in the green tea leaves extracts was determined by high-performance liquid chromatography (HPLC 1260 series, Agilent Technologies, Waldbronn, Germany) with a zorbax eclipse plus C18 column (4.6 × 150 mm, 5 μm pore size, Agilent Technologies). The HPLC solution consisted of purified water, ethanol, and phosphoric acid, and was prepared by mixing at a ratio of 48:50:2 (*v*/*v*/*v*). To produce the analytical samples, 0.025 g of freeze-dried extract was homogeneously dissolved in 5 mL of HPLC solution for 10 min using a vortex mixer. Before the analysis by HPLC, filtering of the analytical samples was performed by a 0.45 μm polyvinylidene fluoride (PVDF) filter (Whatman, UK). Two types of HPLC solvents were used for the analysis of tea catechins. Solvent A was 0.1% phosphoric acid in water and solvent B was methanol (solvent B). The flow rate of the solvent was 1.0 mL/min. The first ratio of solvent A to B was stated from 80 to 20% and gradually changed the ratio of solvent A to 75% for 25 min, then to 65% for 25 min, and finally to 75% for 10 min during the analysis. In this method, the detection wavelength was 280 nm and the detection was conducted by the wavelength detector (Variable Wavelength Detector, Agilent Technologies, PaloAlto, CA, USA). For the analysis, 10 mL were injected from the sample to the HPLC mobile phase.

### 2.6. Data Analysis

The quantitative analysis of six catechins of green tea extracts was carried out through the spiking experiment using the standard of each catechin, and the concentration of catechin (mg/g green tea leaves) was analyzed by the calibration curves of each catechin. The concentration of the calibration range was 50, 100, 200, 500, 1000 and 2000 ppm. The standards of six different tea catechins were purchased from Sigma-Aldrich (St. Louis, MO, USA). The catechin content was expressed as the mean and standard deviation (mean ± SD). All of the extractions were performed three times. The level of significant differences between the concentration of tea catechin according to the PEF and IPL treatment conditions were performed through Duncan’s test (*p* < 0.05) by SPSS statistics (Version 26, IBM, Chicago, IL, USA).

### 2.7. Environmental Scanning Electron Microscopy (ESEM)

The surface morphology of the green tea leaves was observed using the environmental scanning electron microscopy (FEI XL-30FEG, FEI, Burlington, VT, USA) in the Korea Institute of Science and Technology (KIST). The surface of untreated green tea samples and green tea leaves after the pretreatment under all of the PEF and IPL conditions was observed by the magnification of 500× under 0.3–0.7 torr at 25 °C.

## 3. Results

### 3.1. HPLC Analysis of Tea Catechins

The standard of six kinds of catechins was diluted with the HPLC solution to obtain the standard curve of each catechin. The R^2^ values of standard curves were 1 for GCG and Ct, 0.9998 for EGCG, CG, and EC, and 0.9993 for ECG. Appendix A showed the HPLC chromatogram of the green tea leaves extract using subcritical water extraction.

### 3.2. Effects of the Combined Treatment of PEF and SWE

The content of the six kinds of catechins of green tea leaves extracts, which was obtained by the combined treatment of PEF and SWE under various conditions, was shown in Figure 1.

#### 3.2.1. Effects of SWE

The tendency of the extraction content according to the SWE extraction temperature was identified by dividing the content into epicatechins (EGCG, ECG, EC) and their epimers (GCG, CG, Ct) (Figure 1). The optimum extraction temperatures of SWE for EGCG, ECG, and EC were 130, 130, and 110 °C, respectively. The concentration of three kinds of epicatechins in green tea extracts decreased when the extraction temperatures were over 150 °C, and their contents rapidly decreased with the increasing temperature to 190 °C. In contrast, the optimum temperature of SWE for three types of catechins, GCG, CG, and Ct, epimers of each of epicatechin, were all 150 °C. Their concentrations were also decreased when the extraction temperatures were over 150 °C. In general, as the extraction temperature of SWE reached 190 °C, the amount of catechins tended to decrease. It was also confirmed that epicatechins exhibit maximum extraction efficiency at lower extraction temperatures than their epimers. This resulted in the epimerization of epicatechin at 150 °C, which increased the content their epimers.

#### 3.2.2. Effects of PEF

In the case of EGCG, as shown in (Figure 1A), the extracted EGCG was increased until the extraction temperature reached 130 °C. It was also confirmed that the content of EGCG increased by the strong PEF treatment condition at 110 and 130 °C. At the extraction temperature above 150 °C, the content decreased at all of the PEF treatment conditions. Therefore, the optimum extraction condition of EGCG by the combination of PEF and SWE was 3 kV/cm—60 s and 130 °C—5 min, respectively (12.88 ± 0.89 mg/g green tea leaves). In contrast, the contents of GCG (Figure 1B) and ECG (Figure 1C) did not show significant changes between the control (non-PEF treated) and the PEF treatment sample at an optimum condition. The optimum condition of GCG by the combination of PEF and SWE was 0 kV/cm (non-treated) and 150 °C—5 min (13.29 ± 0.15 mg/g green tea leaves). The optimum condition of ECG was 2 kV/cm—60 s of PEF treatment and 130 °C—5 min of SWE (7.10 ± 0.41 mg/g green tea leaves). The extraction content of CG by 2 kV/cm of PEF treatment at 130 and 150 °C was increased significantly. The optimum condition of CG was 2 kV/cm—60 s of PEF treatment and 150 °C—5 min of SWE (3.48 ± 0.17 mg/g green tea leaves) (Figure 1D)). The optimum extraction temperature of CG for non-PEF treated green tea leaves was 170 °C, but when the PEF was treated at 2 kV/cm, the optimum extraction temperature was reduced to 150 °C. EC (Figure 1E) and Ct (Figure 1F) were significantly increased by the PEF treatment of 1 kV/cm. The optimum extraction condition of EC was 1 kV/cm—60 s, 110 °C—5 min (13.36 ± 0.56 mg/g green tea leaves), and 1 kV/cm—60 s, 150 °C—5 min (6.23 ± 0.21 mg/g green tea leaves) for Ct.

From the total catechin concentration of the combined treatment of PEF and SWE, as shown in (Figure 2), it was found that the PEF pretreatment has a positive effect on the subcritical water extraction of tea catechins from the green tea leaves. The optimum condition for the extraction of total catechin was 2 kV/cm—60 s of PEF treatment at 130 °C of the extraction temperature (44.35 ± 2.00 mg/g green tea leaves). It was 15.43% higher than the extraction content of the optimum condition of the control (non-PEF treated) at 150 °C (38.42 ± 0.88 mg/g green tea leaves). Therefore, it was confirmed that the extraction of tea catechin in the green tea leaves damaged by the PEF of appropriate strength was more effective than using the subcritical water alone. The optimum extraction temperature was also reduced from 150 to 130 °C by the PEF pretreatment.

### 3.3. Effects of the Combined Treatment of IPL and SWE

The content of the six kinds of catechins of green tea leaves extracts, which was obtained by the combination of IPL and SWE under various conditions, was shown in Figure 3.

#### 3.3.1. Effects of SWE

Similar to the result of the merge extraction process of PEF and SWE, the tendency of the extraction content with the extraction temperature of SWE could be identified by dividing the content into epicatechins (EGCG, ECG, EC) and their epimers (GCG, CG, Ct) (Figure 3). The optimum temperatures for extracting the EGCG and EC were 110 and 130 °C, respectively. The three kinds of epicatechins in green tea extracts were decreased when the extraction temperatures were above 150 °C, and their contents rapidly decreased with the increasing temperature to 190 °C. In contrast, the three types of catechins, GCG, CG, and Ct, epimers of each of epicatechin, showed optimal extraction conditions at extraction temperatures of 150 °C, and the extraction amount increased up to 110–150 °C, but gradually decreased at higher temperatures above 150 °C.

#### 3.3.2. Effects of IPL

Figure 3A is a graph showing the content of extracted EGCG from green tea leaves with the combined treatment of IPL and SWE. The optimum extraction condition of EGCG by the combination of IPL and SWE was 800 V—60 s, 110 °C—5 min (17.23 ± 1.11 mg/g green tea leaves). In contrast, the contents of GCG (Figure 3B) and ECG (Figure 3C) did not show significant changes between the control and the IPL treatment sample at an optimum condition. The optimum condition of GCG was 1200 V—60 s, 150 °C—5 min (14.82 ± 0.57 mg/g green tea leaves) and 1000 V—60 s, 130 °C—5 min (7.58 ± 0.40 mg/g green tea leaves) for ECG. In Figure 3D, the extraction content of CG was significantly increased by the IPL treatment at 130 and 150 °C. The optimum condition of CG was 1200 V—60 s, 150 °C—5 min (3.64 ± 0.19 mg/g green tea leaves). The optimum extraction temperature for the untreated (non-IPL treated) green tea leaves was 170 °C, but it was decreased to 150 °C by the IPL pretreatment. The optimal merge processing condition of EC was the IPL treatment of 800 V—60 s and SWE of 110 °C—5 min (11.20 ± 0.71 mg/g green tea leaves) (Figure 3E). However, Ct showed no significant difference in the extraction content according to the IPL treatment at an optimum temperature (Figure 3F). The optimum condition of Ct was 800 V—60 s, 150 °C—5 min (5.72 ± 0.34 mg/g green tea leaves).

The total catechin concentration of IPL and SWE combined treatment was shown in Figure 4. It was confirmed that the IPL pretreatment had a positive effect on the improvement of the subcritical water extraction of tea catechins by the concentration of total catechins. The optimum conditions of the combined treatment IPL and SWE were 800 V—60 s and 130 °C—5 min, respectively (48.06 ± 5.03 mg/g dry green tea leaves). This was an increase of about 25.09% from the maximum catechin content of control at 150 °C (38.42 ± 0.88 mg/g green tea leaves). In addition, the results of previous studies showed that SWE has similar or slightly higher extraction efficiency of tea catechins from green tea leaves than extraction using ethanol or methanol as a solvent [11]. Based on the improvement of the extracted catechins concentration by IPL, the merged process with IPL is superior to the conventional organic solvent extraction.

### 3.4. Surface Morphology Analysis

Figure 5A–D shows the surface of green tea leaves treated by various PEF treatment conditions taken using ESEM. Figure 5A shows the surface of untreated green tea leaves. It had a smooth surface compared to the other PEF-treated green tea samples. As the treatment condition became stronger, the surface of the green tea leaves became more uneven and several fragments were formed. In particular, as shown in (Figure 5C), the physical effect of PEF on the surface of green tea leaves was clearly observed from the PEF treatment condition of 2 kV/cm—60 s. In the case of green tea leaves treated with 3 kV/cm—60 s in (Figure 5D), which was the most powerful PEF condition, the surface was destructed and several holes were observed. Figure 5E–H shows the surface of untreated green tea leaves and IPL-treated green tea leaves at various treatment conditions taken using environmental scanning microscopy. Figure 5E shows the surface of untreated green tea leaves. It was smooth compared to the surface of IPL-treated samples.

## 4. Discussion

The results of previous studies showed that the SWE has a similar or slightly higher extraction efficiency of tea catechins from green tea leaves than the extraction using ethanol or methanol as a solvent [11]. Based on the improvement of the extracted catechins concentration by PEF, the merged process with PEF is superior to the conventional organic solvent extraction. PEF applies currents to plant cell membranes, causing them to polarize, then forms temporary pores. At this time, the formed pore was not involved in the movement of substances between the cell membranes [31,32]. However, when PEF was applied to the green tea leaves above the threshold, the electroporation of green tea cells occurs, increasing the extraction efficiency. Moreover, the destruction of vacuoles, which are stored in large quantities of plant active substances, increases the extraction efficiency of the active substances [33]. Furthermore, Wiktor et al. (2015) observed that the total carotenoid content of PEF-treated apple at 1.85 kV/cm increased up to 11.34%, thus they confirmed the electroporation efficiency of PEF [34]. Through these studies, it can be concluded that the total catechin concentration of PEF-treated green tea was higher than the untreated green tea, and 1 kV/cm was higher than the threshold causing the electroporation of green tea. In addition, 2 kV/cm was considered as involved in the electroporation of the vacuole membrane, since it promoted the catechin content to higher than 1 kV/cm. The physical damage, such as cracks and debris on the surface of PEF-treated green tea leaves, was also discussed. When the sample was treated above a strong electric field strength, such as 3 kV/cm, some of the extracted catechins were destroyed by the strong PEF treatment, and the content of catechin was reduced as compared with the treatment at 2 kV/cm. In addition, the strong PEF treatment causes the enzyme-like polyphenol oxidase to leak from the cell. The enzymes react with the extracted catechins, causing the oxidation of catechin to reduce the catechin content [35]. The calculated *Z* value of PEF-treated green tea leaves were 0.08, 0.09, and 0.33 for the PEF treatment of 1, 2, and 3 kV/cm for 60 s, respectively. The *Z* value increased as the treatment conditions of PEF became stronger, and it was confirmed that various electrolytic materials, which are located inside the cell wall, were pulled by the physical destruction and electroporation of the cell by PEF [32].

The extraction tendency has been confirmed in previous studies, where six kinds of catechins of green tea (EGCG, ECG, EC, GCG, CG, and Ct) were extracted at various temperatures using subcritical water extraction [11]. EGCG can be converted into GCG by epimerization in the course of heat application, and fragmentation also can be occurred when the high temperature of heat is applied, resulting in the fragmentation of EGCG or other dimers due to the separation of galloyl group. The galloyl group that separated from EGCG or other catechins became the gallic acid, which is found in leaves, stems, roots, and fruits of various plants and has a good antioxidant ability. In vitro experiments with EGCG confirmed that some EGCGs were converted to GCGs, and some showed that the galloyl group was broken by fragmentation and the content of gallic acid was increased [36]. In another study, the high content of phenolic acid containing a large amount of gallic acid was extracted from the dried red grape skin at a high temperature of SWE [37]. Previous studies have shown that the solvent was more likely to penetrate into the matrix of green tea cells and the solutes were increased by destructing intracellular vacuoles and plant walls. In addition, the solubility was increased by the pressure [38]. Jun et al. (2011) also identified that the ultrahigh pressure extraction (400 MPa) caused the plasma separation of green tea leaves, decreased cell wall thickness, and disrupted organelles through SEM and TEM imaging [39]. Therefore, in this study, it was confirmed that the primary physical destruction was not only by the PEF pretreatment, but also the SWE caused physical damage to the green tea cells through high pressure, which had an effect on maximizing the extraction efficiency of the merging treatment.

IPL facilitates the extraction of functional substances by destroying cell walls or cell membranes of plant cells using pulses of strong light [40]. Hossain et al. (2015) also mentioned that the IPL treatment can induce the aggregation of the cytoplasm, leading to the destruction of the cell membrane, thereby the extraction of a desired substance from the plant becomes easy [33]. Lopes et al. (2016) submitted 0.6 J/cm^2^ of IPL to Tommy Atkins mangoes and they observed that the total antioxidant activity was 130% higher in the IPL-treated pulp and 145% higher in the peel, when compared to the control [41]. In a study by Pataro et al. (2015), immature green tomatoes were exposed to 1–8 J/cm^2^ of IPL or UV-C. They observed that the content of lycopene, total carotenoid, phenolic compounds, and antioxidant activity increased up to 6.2, 2.5, 1.3, and 1.5-fold, respectively, when compared to the control [42]. Therefore, it can be concluded that IPL has a great effect on the enhancement of beneficial food compounds extraction. In this study, IPL showed more effect on the enhancement of extraction of catechin content and a largest increase in EGCG than PEF. This tendency can be discussed by the fact that IPL promotes the synthesis of biosynthetic substances in plants through enhancing the activity of enzymes that are involved in the biosynthesis of metabolites in plants [43,44]. The extraction tendency was similar to the PEF and SWE combined treatment and it was due to the epimerization and degradation of catechin at high temperature of SWE [30,36,37]. In addition, the secondary physical destruction by the SWE pressure had a positive effect on improving the extraction efficiency of the merge process [38,39].

As the PEF treatment conditions increased, the *Z* value and physical destruction observed through the ESEM were simultaneously increased together. Through this study, it was confirmed that permeabilization through the PEF treatment for enhancing the extraction efficiency of tea catechins from green tea leaves and through the microscopic visualization can confirm the degree of plant cell destruction by PEF. In the related studies, SEM images of PEF-treated, fresh, and dry green leaves were observed. Fresh tea leaves have a smooth surface, but after the PEF, cracks and pores were formed on the surface of the leaves. Herein, it was also confirmed that the damage occurred at the protruding part of the surface of the leaves. The surface of the green tea leaves is sufficiently long and acts as a conductor to accumulate perpendicular fields at both ends, resulting in high local electric fields and local breakage, which can release sufficient energy to punch holes [35]. The physical destruction degree of green tea leaves increased as the IPL treatment conditions became stronger, such as the PEF-treated leaves. These results demonstrate that the physical destruction by PEF or IPL results in the fact that higher contents of tea catechins were obtained from SWE from the pretreated green tea leaves than the extracts of untreated green tea leaves. These results are in line with the results of the previous paper, in which quercetin is extracted from onion skin through a combination of IPL and SWE. The surface images of the IPL-treated onion skin formed by ESEM and fluorescence microscopy showed that the damage degree of both images increased at the strong IPL treatment conditions [45].

## 5. Conclusions

The aim of this study is to confirm the effect of PEF and IPL treatment in enhancing the SWE of six kinds of tea catechins from green tea leaves. For the combination of PEF and SWE, the highest concentration of total catechin was 44.35 ± 2.00 mg/g dry green tea leaves at PEF treatment conditions of 2 kV/cm during 60 s, and the SWE temperature of 130 °C. The highest concentration of total catechin was 48.06 ± 5.03 mg/g dry green tea leaves at 800 V during 60 s when the extraction temperature was 130 °C for the combination of IPL and SWE. Compared to the untreated sample, the PEF and IPL treatment significantly enhanced the extraction yield of tea catechins. The total catechin content was increased by 15.43% for PEF and 25.09% for IPL compared to the value of untreated green tea leaves. The results of this study have shown that the PEF or IPL treatment has the potential, as an emerging pretreatment technology, to enhance the SWE of catechins from green tea leaves. Since there are not many studies on non-thermal technologies to improve the extraction yield of functional materials, further studies on pretreatment technologies would contribute to the development of SWE.

In conclusion, SWE is a highly efficient method for recovering valuable catechins from dried green tea leaves. Moreover, the necessity and importance of the research on the merging process of new processing technologies for the development of the extraction of functional materials from various plants were confirmed. The combination of PEF or IPL with SWE makes it possible to extract a large amount of functional materials in a short time by the combined rather than the conventional extraction. In addition, by not using an organic solvent, it is possible to manufacture a high-quality functional material that is environmentally friendly and does not adversely affect the human body. The results presented in this study are useful for setting up a pilot version of the commercial equipment of combined SWE with non-thermal pretreatment technologies.

## Figures and Tables

**Figure 1 foods-10-03092-f001:**
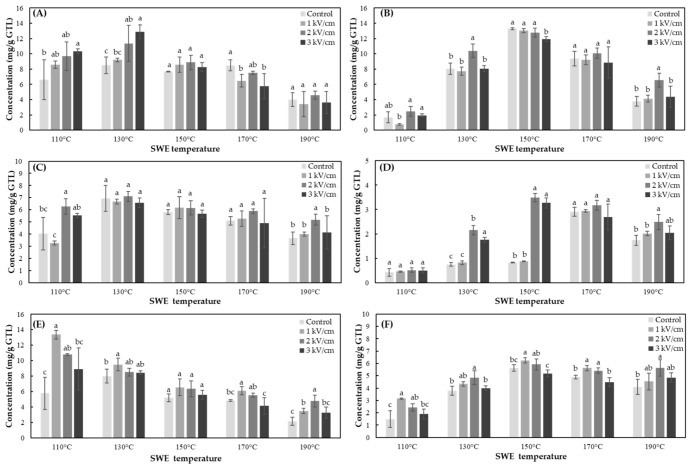
Concentration of six types of catechins of PEF pretreated green tea extracts using SWE. (**A**) EGCG, (**B**) GCG, (**C**) ECG, (**D**) CG, (**E**) EC, and (**F**) Ct. Treatment times of SWE and PEF are 5 min and 60 s, respectively. Data are mean and SD values. The lowercase letters a, b, and c indicate the level of significant differences performed using Duncan’s test (*p* < 0.05).

**Figure 2 foods-10-03092-f002:**
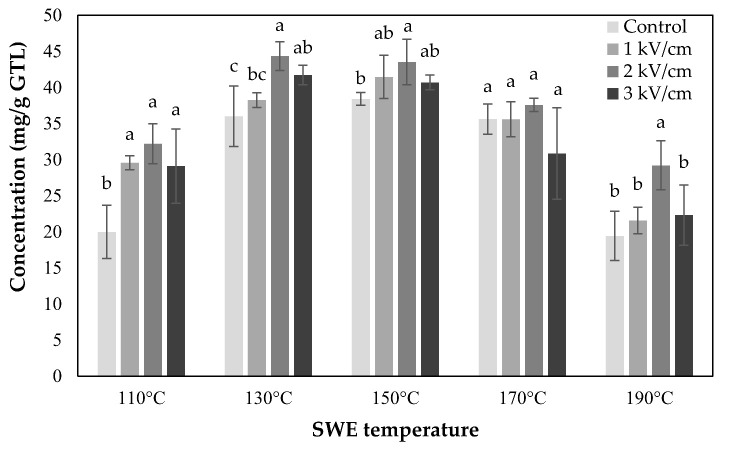
Total catechin concentration of PEF pretreated green tea extracts using SWE. Treatment times of SWE and PEF are 5 min and 60 s, respectively. Data are mean and SD values. The lowercase letters a, b, and c indicate the level of significant differences performed using Duncan’s test (*p* < 0.05).

**Figure 3 foods-10-03092-f003:**
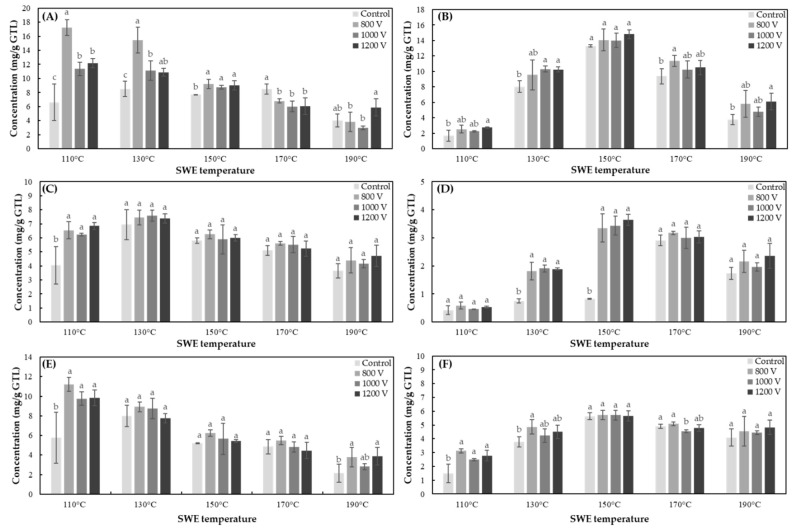
Concentration of six types of catechins of IPL pretreated green tea extracts using SWE. (**A**) EGCG, (**B**) GCG, (**C**) ECG, (**D**) CG, (**E**) EC, and (**F**) Ct. Treatment times of SWE and IPL are 5 min and 60 s, respectively. Data are mean and SD values. The lowercase letters a, b, and c indicate the level of significant differences performed using Duncan’s test (*p* < 0.05).

**Figure 4 foods-10-03092-f004:**
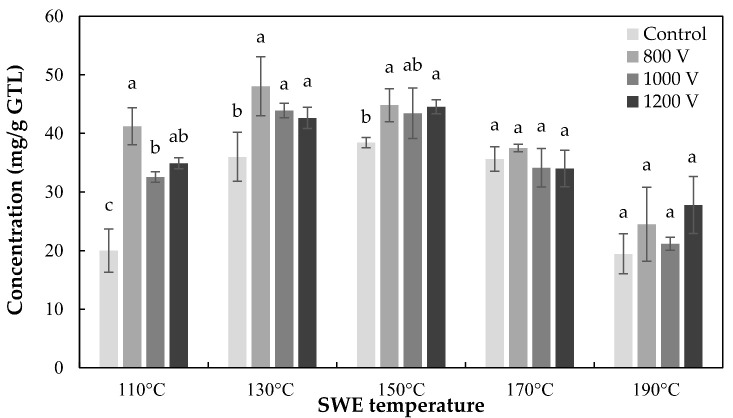
Total catechin concentration of IPL pretreated green tea extracts using SWE. Treatment times of SWE and IPL are 5 min and 60 s, respectively. Data are mean and SD values. The lowercase letters a, b, and c indicate the level of significant differences performed using Duncan’s test (*p* < 0.05).

**Figure 5 foods-10-03092-f005:**
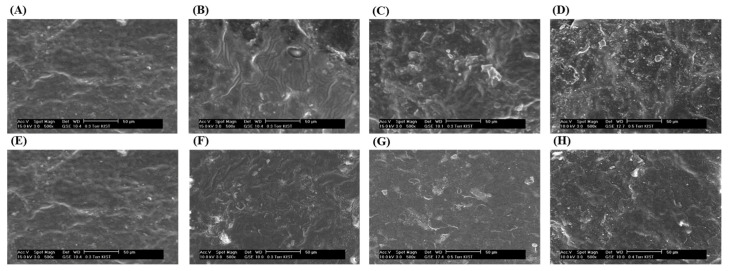
ESEM of green tea leaves after 60 s of PEF and IPL pretreatment magnified to 500×. (**A**) PEF-Control, (**B**) PEF-1 kV/cm, (**C**) PEF-2 kV/cm, (**D**) PEF-3 kV/cm, (**E**) IPL-Control, (**F**) IPL-800 V, (**G**) IPL-1000 V, and (**H**) IPL-1200 V.

## Data Availability

The datasets generated for this study are available on request from the corresponding author.

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
