# Peer review of "Improving the Extraction of Catechins of Green Tea (Camellia sinensis) by Subcritical Water Extraction (SWE) Combined with Pulsed Electric Field (PEF) or Intense Pulsed Light (IPL) Pretreatment"

_foods, 2021, doi:10.3390/foods10123092_

Round 1

Reviewer 1 Report

Authors still have not merged figures back into the main text. It is inconvenient to refer back to supplementary files all the time. Some improvements have been made from the previous version. 

Author Response

I really appreciate your useful comments and suggestions on our manuscript. We have revised the manuscript accordingly, and detailed changes and explanations are listed below point by point:

Reviewer 1: Authors still have not merged figures back into the main text. It is inconvenient to refer back to supplementary files all the time. Some improvements have been made from the previous version. (Answer) As the reviewer suggested, Figures 3–7 have been inserted into the manuscript.

The manuscript has been resubmitted to your journal. We look forward to your positive response.

Thank you.

Reviewer 2 Report

Present research by Hwang et al. is focused on the improvement of subcritical water extraction (SWE) of green tea catechins by application of pulsed electric field (PEF) and intense pulsed light (IPL) pretreatments. Green tea catechins are very important group of compounds and this research is focused on combination of novel and emerging techniques which achieved improvement in catechnins yield. Paper should be interesting for the Foods audience, however, i have some important remarks and comments which should be accessed by the authors. My comments are given in attached pdf file.

Author Response

I really appreciate your useful comments and suggestions on our manuscript. We have revised the manuscript accordingly, and detailed changes and explanations are listed below point by point:

Reviewer 2: Present research by Hwang et al. is focused on the improvement of subcritical water extraction (SWE) of green tea catechins by application of pulsed electric field (PEF) and intense pulsed light (IPL) pretreatments. Green tea catechins are very important group of compounds and this research is focused on combination of novel and emerging techniques which achieved improvement in catechnins yield. Paper should be interesting for the Foods audience, however, I have some important remarks and comments which should be accessed by the authors.

L4. add "pretreatment" as the last word

(Answer) The word has been added in the manuscript (line 4).

L29-30. Even though green tea polyphenols have exhibited admirable bioactive potential, some limitations of these compounds (such as poor bioavailability) must be highlighted.

(Answer) The content has been added in the manuscript as the reviewer suggested (lines 47-51).

L53. and moderately polar

(Answer) This has been added in the manuscript (line 56).

L55. Express temperature in C or K, be consistent.

(Answer) Temperature has been written in °C (line 58).

L67. be more specific, what is the temperature?

(Answer) The specific temperature have been written (line 71).

L93. Where there any successful studies with combining effects of SWE, PEF and IPL on some plant matrices? Please add and cite.

(Answer) There are some studies combining the effects of general organic solvents like ethanol and methanol with PEF/IPL, and they are written in lines 76-84 and 87-91. But the studies applying for SWE as an extraction method are very rare.

L102. When? Where exactly, add geographical ordinates?

(Answer) We purchased green tea leaves product, which are known to be cultivated in Boseong, Korea, and we do not know exactly when and where the leaves are grown and harvested.

L103. How was drying performed? What was the particle size?

(Answer) The manuscript has been revised (106). And the particle size of green tea leaves was very diverse, so the experiment was conducted by determining a uniform weight of dried green tea leaves for each treatment (IPL, PEF, SWE). These weights are written in each session (lines 113, 132, 146).

L147-148. What about other extraction parameters? How did you select the extraction time? What was applied pressure? What about rinse volume and the cycles applied on ASE?

(Answer) The treated temperature was set based on the capacity of the equipment and the conditions of previous research (Ko, M.J.; Cheigh, C.I.; Chung, M.S. Optimization of subcritical water extraction of flavanols from green tea leaves. J. Agric. Food. Chem. 2014, 62, 6828-6833.). This information has been written in the manuscript (lines 149-151).

L175. Please add the concentrations of the calibration range.

(Answer) The concentrations of the calibration range has been added (lines 177-178).

L196. I don't understand why did you put all figures in supplementary data. At least figures S3-S6 should be directly in the paper! Also, something is wrong with the whole figure 3. Y-axis is not correct since it is the same for all figures, please check and correct.

(Answer) Figures 3–7 have been inserted into the manuscript and they have been revised as the reviewer pointed out.

L198. You should delete extraction time from the X-axis in all figures since it was always 5 min!

(Answer) The figures have been revised as the reviewer pointed out.

L227. Results and discussion should start with effects of SWE, without any other pretreatment and after that you could compare effects, please correct.

(Answer) The sections have been revised as the reviewer suggested (section 3.2.1–3.2.2 and 3.3.1–3.3.2)

L243. Same comments as it was in the figure 3, please check and correct x and y axis.

(Answer) The figures have been revised as the reviewer pointed out.

L297. Discussion is missing on energy consumption and environmental effect of each technique. Since you did not measure anything in the results, you should at least give some explanations with appropriate references.

(Answer) The energy consumption or environmental effect were not studied in our study. This information is written in lines 48-51 and 92-95.

L404-405. Please give suggestions for potential application of these results and extracts.

(Answer) The contents of conclusion have been complemented (lines 422-431).

The manuscript has been resubmitted to your journal. We look forward to your positive response.

Thank you.

Reviewer 3 Report

The article titled: "Improving the extraction of catechins of green tea (Camellia sinensis) by subcritical water extraction (SWE) combined with pulsed electric field (PEF) or intense pulsed light (IPL)"  deals with the apply of pulsed electric field (PEF) and intense pulsed light (IPL) for the enhancement of subcritical water extraction (SWE) of tea catechins from green tea leaves. The subject of the article is interesting, but it have some mistakes in the text. The comments and recommendations of the Reviewer are listed below:

  1. There are a lot of mistakes in sentences regarding the style and grammar, so the text should be read very carefully to correct all the mistakes.
  2. line 29 - it should be: "because of their attractive aroma" instead of: "its..."
  3. line 29 - it should be: "Because of their excellent health..."
  4. line 31 - it should be: "have been conducted"
  5. line 31 - "their abilities"
  6. line 37: "exist"
  7. line 66 - it should be: "has shown"
  8. lines 73-76 - the sentence is incorrect: "In many studies improved....conditions for improving" - please rewrite it.
  9. lines 88-89 - "also" is used for two times - please rewrite the sentence.
  10. lines 103 - it should be: "were purchased"
  11. lines 105-107 - this sentence does not belong to "Materials and methods", but "Results" - please remove it from this section.
  12. line 186 - it should be: "was observed"
  13. line 196 - "in Figure 3S"
  14. line 204 - what does it mean: "the content of GCG were not able to observe significant changes..." - the style is incorrect, please rewrite the sentence.
  15. lines 206-208 - the style is incorrect, please rewrite the sentence.
  16. lines 208-210 - the style is incorrect, please rewrite the sentence.
  17. line 218 - "has a positive effect"
  18. line 224 "was more effective than..."
  19. line 229 - it should be: "was".
  20. lines 230-231 - what does it mean: "EGCG and ECG has optimum conditions" - the style is incorrect, please rewrite the sentence.
  21. lines 231-232 - "Epikatechin began to decrease" - what does it mean? The style is incorrect and the sentence should be rewritten.
  22. lines 233-234 - "catechins...had optimal extraction conditions" - the style is incorrect, please rewrite the sentence.
  23. line 235 - extraction amount increased?
  24. lines 248-249 - "the contents ...were not able to observe"? - the style is incorrect, please rewrite the sentence.
  25. line 256 - "EC was significantly increased by..." - please correct the style of the sentence.
  26. lines 276-278- the sentence is not correct. Please, rearrange it.
  27. line 278 -"epicatechins began to decrease"? - the style is to be corrected
  28. line 281 - "had optimal extraction conditions" (?)
  29. line 286 - "that" should be removed.
  30. lines 340, 347, 349, 352 - names of the Authors should be given before brackets.
  31. lines 365-368 - the sentence is incomprehensible - please rewrite it.
  32. line 398 - the process of decontamination of microorganism has not been studied by the Authors, so it should not be mentioned in Conclusions  section.
  33. lines 398-400 - the sentence is incomprehensible.
  34. Please, present all the results in tables in the text of the article.
  35. There are no figures presented in the text. All figures are shown in suplementary files. The most important figures should be incorporated into main text of the article.

Author Response

I really appreciate your useful comments and suggestions on our manuscript. We have revised the manuscript accordingly, and detailed changes and explanations are listed below point by point:

Reviewer 3: The article titled: "Improving the extraction of catechins of green tea (Camellia sinensis) by subcritical water extraction (SWE) combined with pulsed electric field (PEF) or intense pulsed light (IPL)" deals with the apply of pulsed electric field (PEF) and intense pulsed light (IPL) for the enhancement of subcritical water extraction (SWE) of tea catechins from green tea leaves. The subject of the article is interesting, but it have some mistakes in the text. The comments and recommendations of the Reviewer are listed below: There are a lot of mistakes in sentences regarding the style and grammar, so the text should be read very carefully to correct all the mistakes.

The whole manuscript was edited by a professional English editor. (English Science Editing (english-science-editing.com))

line 29 - it should be: "because of their attractive aroma" instead of: "its..."

(Answer) This has been revised as the reviewer suggested (line 30).

line 29 - it should be: "Because of their excellent health..."

(Answer) This has been revised as the reviewer suggested (line 30).

line 31 - it should be: "have been conducted"

(Answer) This has been revised as the reviewer suggested (line 32).

line 31 - "their abilities"

(Answer) This has been revised as the reviewer suggested (line 32).

line 37: "exist"

(Answer) This has been revised as the reviewer suggested (line 38).

line 66 - it should be: "has shown"

(Answer) This has been revised as the reviewer suggested (line 70).

lines 73-76 - the sentence is incorrect: "In many studies improved....conditions for improving" - please rewrite it.

(Answer) The sentence has been revised (lines 76-77).

lines 88-89 - "also" is used for two times - please rewrite the sentence.

(Answer) The sentence has been revised as the reviewer suggested (lines 91-92).

lines 103 - it should be: "were purchased"

(Answer) This has been revised as the reviewer suggested (line 106).

lines 105-107 - this sentence does not belong to "Materials and methods", but "Results" - please remove it from this section.

(Answer) This has been written in lines 224 and 282.

line 186 - it should be: "was observed"

(Answer) This has been revised as the reviewer suggested (line 188).

line 196 - "in Figure 3S"

(Answer) This has been revised as the reviewer suggested (line 198).

line 204 - what does it mean: "the content of GCG were not able to observe significant changes..." - the style is incorrect, please rewrite the sentence.

(Answer) The sentence has been revised as the reviewer suggested (line 223).

lines 206-208 - the style is incorrect, please rewrite the sentence.

(Answer) The sentence has been revised as the reviewer suggested (lines 225-228).

lines 208-210 - the style is incorrect, please rewrite the sentence.

(Answer) The sentence has been revised as the reviewer suggested (lines 228-231).

line 218 - "has a positive effect"

(Answer) This has been revised as the reviewer suggested (line 238).

line 224 "was more effective than..."

(Answer) This has been revised as the reviewer suggested (line 224).

line 229 - it should be: "was".

(Answer) This has been revised as the reviewer suggested (line 203).

lines 230-231 - what does it mean: "EGCG and ECG has optimum conditions" - the style is incorrect, please rewrite the sentence.

(Answer) The sentence has been revised as the reviewer suggested (lines 205-208).

lines 231-232 - "Epicatechin began to decrease" - what does it mean? The style is incorrect and the sentence should be rewritten.

(Answer) The sentence has been revised as the reviewer suggested (lines 206-208).

lines 233-234 - "catechins...had optimal extraction conditions" - the style is incorrect, please rewrite the sentence.

(Answer) The sentence has been revised as the reviewer suggested (lines 208-211).

line 235 - extraction amount increased?

(Answer) The sentence has been revised as the reviewer suggested (lines 208-211).

lines 248-249 - "the contents ...were not able to observe"? - the style is incorrect, please rewrite the sentence.

(Answer) The sentence has been revised as the reviewer suggested (lines 273-274).

line 256 - "EC was significantly increased by..." - please correct the style of the sentence.

(Answer) The sentence has been revised as the reviewer suggested (lines 281-283).

lines 276-278- the sentence is not correct. Please, rearrange it.

(Answer) The sentence has been revised as the reviewer suggested (lines 261-262).

line 278 -"epicatechins began to decrease"? - the style is to be corrected

(Answer) The sentence has been revised as the reviewer suggested (lines 263-264).

line 281 - "had optimal extraction conditions" (?)

(Answer) The sentence has been revised as the reviewer suggested (lines 266).

line 286 - "that" should be removed.

(Answer) This has been removed as the reviewer suggested (lines 305-306).

lines 340, 347, 349, 352 - names of the Authors should be given before brackets.

(Answer) The names of the authors have been written in the manuscript (lines 362,370,373,375).

lines 365-368 - the sentence is incomprehensible - please rewrite it.

(Answer) The sentence has been deleted because it did not fit the contents of the discussion.

line 398 - the process of decontamination of microorganism has not been studied by the Authors, so it should not be mentioned in Conclusions  section.

(Answer) The sentence has been revised as the reviewer pointed out (lines 419-421).

lines 398-400 - the sentence is incomprehensible.

(Answer) The sentence has been revised as the reviewer pointed out (lines 419-421).

Please, present all the results in tables in the text of the article.

(Answer) Please understand that the results are presented in figures for easy comparison. The figures have been added to the manuscript.

There are no figures presented in the text. All figures are shown in supplementary files. The most important figures should be incorporated into main text of the article.

(Answer) As the reviewer suggested, Figures 3–7 have been inserted into the manuscript.

The manuscript has been resubmitted to your journal. We look forward to your positive response.

Thank you.

Round 2

Reviewer 2 Report

Authors improved the paper according to my suggestions.

Reviewer 3 Report

I would like to thank the Authors for improving the manuscript according to Reviewer's comments and recommendations.

In my opinion the article is suitable to be published after minor revision. Comments of the Reviewer are listed below:

1. Please correct the sentence in lines 76-77 - it should be: "In many studies it was mentioned that..."

2. line 428 -it should be: "shorter time..." - please correct the sentence

3. Abbreviations used in figures' titles should be explained under figures, that they can be analyzed independently.

Round 1

Reviewer 1 Report

This manuscript reports a subcritical water extraction (SWE) approach combined with extraction processing of pulsed electric field (PEF) or intense pulsed light (IPL) for extracting catechins of green tea. However, the parameters design of test and the results analysis in the manuscript are difficult to understand, especially in the parameters selection of IPL(various voltage (800, 1000, , 1200 V and time of 60 seconds. There are lots of mistakes in orthography, grammar, format for the citations and Figures.

  1. Title:the title does not agree with the content of manuscript. “Improving the extraction of catechins of green tea (Camellia sinensis) by subcritical water extraction (SWE) combined with pulsed electric field (PEF) or intense pulsed light (IPL)” may be better.
  2. Abstract: lines 11-12: pulsed electric field (PEF) and intense pulsed light( IPL)ï¼›Line: delete the word of “extractions”
  3. Keywords: delete abbreviations
  4. Line 100: 2.1. Green tea sample, catechins content of thesample should be added in this part.
  5. Figure:FigureS3.-Figure 6: the appearance of this figures are difficult to understand, Does all vertical axis display catechins content? There's no annotations about the lowercase letters (a,b) in those figures. In addition, differences between groups should be shown.
  6. In Figure S6, authors presented that most data of catechins content are no difference by subcritical water extraction combined with pulsed electric field. Why is the yield of catechin so low? I think there may be something wrong about the design of the experiment. What happens if the ranges of voltage or the time are improved? In this part, Intergroup analysis is necessary.
  7. Line 241: “3.3.1. Effects of PEF” need to correct“Effects of IPL”.
  8. Lines 73-80, 243-245, 305: there are mistakes in the gramma and format of citations.
  9. Conclusions: the conclusion is more like a summary, but not a conclusion.

Author Response

Responses to reviewers’ comments

Manuscript ID: foods-1432186

Title: Improving the subcritical water extraction (SWE) of tea catechins from green tea (Camellia sinensis) by pulsed electric field (PEF) and intense pulsed light (IPL)

I really appreciate your useful comments and suggestions on our manuscript. We have revised the manuscript accordingly, and detailed changes and explanations are listed below point by point:

Reviewer 1: This manuscript reports a subcritical water extraction (SWE) approach combined with extraction processing of pulsed electric field (PEF) or intense pulsed light (IPL) for extracting catechins of green tea. However, the parameters design of test and the results analysis in the manuscript are difficult to understand, especially in the parameters selection of IPL (various voltage (800, 1000, 1200 V and time of 60 seconds. There are lots of mistakes in orthography, grammar, format for the citations and Figures.

(Answer) The whole manuscript was edited by a professional English editor. (English Science Editing (english-science-editing.com))

  1. Title:the title does not agree with the content of manuscript. “Improving the extraction of catechins of green tea (Camellia sinensis) by subcritical water extraction (SWE) combined with pulsed electric field (PEF) or intense pulsed light (IPL)” may be better.

(Answer) The title has been changed as the reviewer suggested.

  1. Abstract: lines 11-12: pulsed electric field (PEF) and intense pulsed light(IPL)ï¼›Line: delete the word of “extractions”

(Answer) These have been revised (lines 11-12, 15-16).

  1. Keywords: delete abbreviations

(Answer) The abbreviations have been deleted.

  1. Line 100: 2.1. Green tea sample, catechins content of the sample should be added in this part.

(Answer) Catechin content of the sample achieved using SWE at 150°C and 5 min has been written in the manuscript as the reviewer suggested (lines 105-107).

  1. Figure:FigureS3.-Figure 6: the appearance of this figures are difficult to understand, Does all vertical axis display catechins content? There's no annotations about the lowercase letters (a,b) in those figures. In addition, differences between groups should be shown.

(Answer) Figure caption file had been submitted separately. The information about the lowercase letters has been added. Figure captions of Fig.3–6 is as follows;

Fig. 3. Concentration of six types of catechins of PEF pre-treated green tea extracts by using SWE. (A) EGCG, (B) GCG, (C) ECG, (D) CG, (E) EC, and (F) Ct. Data are mean and SD values. The lowercase letters a, b, and c indicate the level of significant differences performed using Duncan's test (p < 0.05).

Fig. 4. Total catechin concentration of PEF pre-treated green tea extracts by using SWE. Data are mean and SD values. Data are mean and SD values. The lowercase letters a, b, and c indicate the level of significant differences performed using Duncan's test (p < 0.05).

Fig. 5. Concentration of six types of catechins of IPL pre-treated green tea extracts by using SWE. (A) EGCG, (B) GCG, (C) ECG, (D) CG, (E) EC, and (F) Ct. Data are mean and SD values. Data are mean and SD values. The lowercase letters a, b, and c indicate the level of significant differences performed using Duncan's test (p < 0.05).

Fig. 6. Total catechin concentration of IPL pre-treated green tea extracts by using SWE. Data are mean and SD values. Data are mean and SD values. The lowercase letters a, b, and c indicate the level of significant differences performed using Duncan's test (p < 0.05).

  1. In Figure S6,authors presented that most data of catechins content are no difference by subcritical water extraction combined with pulsed electric field. Why is the yield of catechin so low? I think there may be something wrong about the design of the experiment. What happens if the ranges of voltage or the time are improved? In this part, Intergroup analysis is necessary.

(Answer) As described in section 3.2.1 of our manuscript, the optimum condition for extraction of total catechin was 2 kV/cm – 60 sec of PEF treatment at 130°C of extraction temperature (44.35 ± 2.00 mg/g green tea leaves). Above 130°C, the extraction yield decreased although the intensity of PEF increased. The results indicate that a range of appropriate PEF strength exists for extracting catechins from the green tea leaves effectively (These were mentioned in lines 217-226)

  1. Line 241: “3.3.1. Effects of PEF” need to correct “Effects of IPL”.

(Answer) The word has been changed (line 244).

  1. Lines 73-80, 243-245, 305: there are mistakes in the gramma and format of citations.

(Answer) The author name and year have been added and the sentence has been revised (lines 76, 79-81, 246-248).

  1. Conclusions: the conclusion is more like a summary, but not a conclusion.

(Answer) Conclusion has been revised as the reviewer suggested (lines 395-401).

The manuscript has been resubmitted to your journal. We look forward to your positive response.

Thank you.

Reviewer 2 Report

Authors have presented application of IPL and PEF in combination with SWE. It was interesting to see application of IPL in green tea. The study should have taken other measurements to characterize the green tea better rather than measuring the amounts of 6 different catechins only. It was unclear whether quantitative or qualitative approach was taken for HPLC analysis and repeats for HPLC analysis were not mentioned. A lot of details were omitted for starting materials and chemicals. Rather than presenting all of the data in figures, some could have been better in tables. It was unclear why figures and tables were all put in as supplementary materials, and not included in the main manuscript. They should be included in the main text instead. The extracted catechins should have been tested for their antioxidant activities and bioavailability. English check is highly recommended - some sentences were hard to understand.

Line 53: non-polar

Line 56-7: research

59-60: revise the sentence

70-71: revise the sentence

76, 78: give author name and year

78-80: revise the sentence

81-91: give details of IPL treatments done to provide such impacts

92-98: was this the first attempt in the literature to apply SWE with PEF/IPL?

98: deducted?

101-104: more details desired. Eg. Growth conditions, environmental conditions, % relative humidity, how was light blocked, what enzymes there were to inactivate, how they were steamed and dried, using what facilities, how long at what temperature?

109: how dried was it, in terms of RH?

125: if self designed, how do you validate it? Was there a preliminary trial to see if IPL was working correctly? How does this differ from IPL used by dermatologists or skin care places?

145: how were those temperatures decided?

150: what were the 6 catechins? What standards and grades were used for chemicals? Repeats for HPLC?

170: why were the 6 catechins not quantified? What can you provide to ensure that the qualitative results are accurate?

187-190: Earlier it said qualitative analysis was done, but in here, R2 values are given. How were those R2 values calculated? Was there one solution standard which had 6 catechins dissolved in it, and that was used for calibration? It’s unclear

Line 241: supposed to be IPL not PEF?

330: italics for in vitro

Figure S7: use annotation to highlight the important parts of the SEM photos

Author Response

Responses to reviewers’ comments

Manuscript ID: foods-1432186

Title: Improving the subcritical water extraction (SWE) of tea catechins from green tea (Camellia sinensis) by pulsed electric field (PEF) and intense pulsed light (IPL)

I really appreciate your useful comments and suggestions on our manuscript. We have revised the manuscript accordingly, and detailed changes and explanations are listed below point by point:

Reviewer 2: Authors have presented application of IPL and PEF in combination with SWE. It was interesting to see application of IPL in green tea. The study should have taken other measurements to characterize the green tea better rather than measuring the amounts of 6 different catechins only. It was unclear whether quantitative or qualitative approach was taken for HPLC analysis and repeats for HPLC analysis were not mentioned. A lot of details were omitted for starting materials and chemicals. Rather than presenting all of the data in figures, some could have been better in tables. It was unclear why figures and tables were all put in as supplementary materials, and not included in the main manuscript. They should be included in the main text instead. The extracted catechins should have been tested for their antioxidant activities and bioavailability. English check is highly recommended - some sentences were hard to understand.

(Answer) The whole manuscript was edited by a professional English editor. (English Science Editing (english-science-editing.com))

  1. Line 53: non-polar

(Answer) The word has been revised (line 53).

  1. Line 56-7: research

(Answer) The word has been revised (line 56).

  1. 59-60: revise the sentence

(Answer) The sentence has been revised (line 59-60).

  1. 70-71: revise the sentence

(Answer) The sentence has been revised (line 70-71).

  1. 76, 78: give author name and year

(Answer) The author name and year has been added (lines 76, 79)

  1. 78-80: revise the sentence

(Answer) The sentence has been revised (lines 79-81).

  1. 81-91: give details of IPL treatments done to provide such impacts

(Answer) The fluence of IPL has been written as the reviewer suggested (lines 87, 89).

  1. 92-98: was this the first attempt in the literature to apply SWE with PEF/IPL?

(Answer) As described in lines 71-72 and 82-83, PEF and IPL are originally developed as non-thermal disinfection technologies. Therefore, studies that applied PEF or IPL to increase the extraction yield of functional substances present in food are not enough. In particular, studies applying for SWE as an extraction method are very rarely.

  1. 98: deducted?

(Answer) The word has been revised (line 99).

  1. 101-104: more details desired. Eg. Growth conditions, environmental conditions, % relative humidity, how was light blocked, what enzymes there were to inactivate, how they were steamed and dried, using what facilities, how long at what temperature?

(Answer) The section 2.1. has been revised entirely (lines 102-107).

  1. 109: how dried was it, in terms of RH?

(Answer) The information of moisture content of dried green tea leaves has been added in the manuscript (line 103).

  1. 125: if self-designed, how do you validate it? Was there a preliminary trial to see if IPL was working correctly? How does this differ from IPL used by dermatologists or skin care places?

(Answer) First of all, the principle of IPL used by skincare places and used by the food industry is the same. Both cases use intense light with a wide range of wavelengths. Depending on wavelength of light, the IPL can destroy the melamine pigment (>590 nm) or sterilize microorganisms (280-400 nm). All researches applying for the IPL to the foods have used self-designed devices which composed of lamp and chamber. The IPL device used in this study was also confirmed to work correctly through many of our previous studies.

  • Hwang, H. J., Park, J. Y., Chung, M. S., & Cheigh, C. I. (2021) Microbial inactivation in fresh and minimally processed foods by intense pulsed light (IPL) treatment. Food Science and Biotechnology, 30(7), 939-948.
  • Hwang, H. J., Kim, G. A., & Chung, M. S. (2021) Impact of factors affecting the efficacy of intense pulsed light for reducing Bacillus subtilis spores. Food Science and Biotechnology, 30(10), 1321-1329.
  • Hwang, H. J., Kim, H. J., Ko, M. J., & Chung, M. S. (2021) Recovery of hesperidin and narirutin from waste Citrus unshiu peel using subcritical water extraction aided by pulsed electric field treatment. Food Science and Biotechnology, 30(2), 217-226.
  • Jo, H. L., Hwang, H. J., & Chung, M. S. (2019) Inactivation of Bacillus subtilis spores at various germination and outgrowth stages using intense pulsed light. Food Microbiology. 82(1): 409-415
  • Hwang, H. J., Seo, J. H., Jeong, C. M., Cheigh, C. I., & Chung, M. S. (2019) Analysis of microorganism inactivation by intense pulsed light using a double-Weibull survival model. Innovative Food Science Emerging Technology. 56: 102185
  • Hwang, H. J., Cheigh, C. I., & Chung, M. S. (2019) Comparison of bactericidal effects of two types of pilot-scale intense-pulsed-light devices on cassia seeds and glutinous millet. Innovative Food Science & Emerging Technology. 49: 170-175
  1. 145: how were those temperatures decided?

(Answer) The treated temperature was set based on the capacity of the equipment and the conditions of previous research (Ko, M.J.; Cheigh, C.I.; Chung, M.S. Optimization of subcritical water extraction of flavanols from green tea leaves. J. Agric. Food. Chem. 2014, 62, 6828-6833.)

  1. 150: what were the 6 catechins? What standards and grades were used for chemicals? Repeats for HPLC?

(Answer) The 6 catechines have been added in this section. As mentioned in lines 154-156, we used the same method and standards as in the previous study.

  1. 170: why were the 6 catechins not quantified? What can you provide to ensure that the qualitative results are accurate?

(Answer) The materials and methods of quantitative analysis is described in line 156–171 and the accuracy was confirmed by our previous study (ref. 30).

  1. 187-190: Earlier it said qualitative analysis was done, but in here, R2 values are given. How were those R2 values calculated? Was there one solution standard which had 6 catechins dissolved in it, and that was used for calibration? It’s unclear.

(Answer) We performed quantitative analysis, it is described in line 156–171, a mistake was corrected in line 173.

  1. Line 241: supposed to be IPL not PEF?

(Answer) The word has been revised (line 244).

  1. 330: italics for in vitro

(Answer) The word has been changed to italics (line 333).

  1. Figure S7: use annotation to highlight the important parts of the SEM photos.

(Answer) We just want to show that the overall surface is rough compared to the control sample.

The manuscript has been resubmitted to your journal. We look forward to your positive response.

Thank you.